# Stimulating the Antitumor Immune Response Using Immunocytokines: A Preclinical and Clinical Overview

**DOI:** 10.3390/pharmaceutics16080974

**Published:** 2024-07-24

**Authors:** Bart Boersma, Hélène Poinot, Aurélien Pommier

**Affiliations:** 1Institute of Pharmaceutical Sciences of Western Switzerland, University of Geneva, 1211 Geneva, Switzerland; bart.boersma@unige.ch; 2School of Pharmaceutical Sciences, University of Geneva, 1211 Geneva, Switzerland; 3Department of Cell Physiology and Metabolism, Faculty of Medicine, University of Geneva, 1211 Geneva, Switzerland; helene.poinot@unige.ch; 4Translational Research Centre in Oncohaematology, University of Geneva, 1211 Geneva, Switzerland; 5UMR1240 Imagerie Moléculaire et Stratégies Théranostiques INSERM, Université Clermont Auvergne, BP 184, F-63005 Clermont-Ferrand, France

**Keywords:** immunocytokines, cytokines, cancer, immune response, immunotherapy

## Abstract

Cytokines are immune modulators which can enhance the immune response and have been proven to be an effective class of immunotherapy. Nevertheless, the clinical use of cytokines in cancer treatment has faced several challenges associated with poor pharmacokinetic properties and the occurrence of adverse effects. Immunocytokines (ICKs) have emerged as a promising approach to overcome the pharmacological limitations observed with cytokines. ICKs are fusion proteins designed to deliver cytokines in the tumor microenvironment by taking advantage of the stability and specificity of immunoglobulin-based scaffolds. Several technological approaches have been developed. This review focuses on ICKs designed with the most impactful cytokines in the cancer field: IL-2, TNFα, IL-10, IL-12, IL-15, IL-21, IFNγ, GM-CSF, and IFNα. An overview of the pharmacological effects of the naked cytokines and ICKs tested for cancer therapy is detailed. A particular emphasis is given on the immunomodulatory effects of ICKs associated with their technological design. In conclusion, this review highlights active ways of development of ICKs. Their already promising results observed in clinical trials are likely to be improved with the advances in targeting technologies such as cytokine/linker engineering and the design of multispecific antibodies with tumor targeting and immunostimulatory functional properties.

## 1. Introduction

Cytokines represent a class of low-molecular-weight proteins secreted by various cell types, including immune cells, endothelial cells, and stromal cells [1]. Cytokines bind to specific receptors on target cells, triggering a variety of intracellular signaling pathways that ultimately govern gene expression and cellular behavior. Cytokines play a pivotal role in orchestrating immune responses, modulating cell differentiation, and facilitating intercellular communication within biological systems [2]. They serve as indispensable regulators of the immune system and play important roles in numerous diseases, including cancer, autoimmune disorders, infectious diseases, and inflammatory conditions [1,2]. 

The genesis of cytokine research traces back to the early 20th century when investigators unveiled the presence of a factor within leukocytes capable of inducing the proliferation and differentiation of neighboring cells [1]. It was not until the 1950s and 1960s that a series of experiments revealed the existence of an intricate network of signaling molecules finely regulating immune responses [3]. This led to the identification of the first cytokine, interleukin(IL)-1, in 1979 [4]. Since then, the field of cytokine research has exploded, with over one hundred cytokines and their receptors identified and characterized [1]. Cytokines have been systematically categorized into various families, predicated on shared structural and functional attributes, with prominent examples encompassing the interleukin (IL), tumor necrosis factor (TNF), interferon (IFN), and chemokine families [1]. As potent immune modulators, cytokines can have pathological implications in certain disorders such as cancer, where their dysregulated production or action can drive tumor growth and impede the immune system’s efficacy in eradicating malignant cells [1]. The immune cell population within the tumor microenvironment (TME) can be categorized into two distinct subtypes: pro-inflammatory and anti-inflammatory cells, as illustrated in Figure 1. Pro-inflammatory cells include CD8+ T cells, M1 macrophages, natural killer (NK) cells, and dendritic cells (DCs). Among these, macrophages, NK cells and CD8+ T cells exhibit the capability to directly engage with tumor cells and eliminate them [5,6]. DCs, macrophages, and neutrophils exert their tumor cell-killing activity by releasing cytokines such as TNFα, IL-6, and IFNγ [7,8,9]. Neutrophils also exert their antitumor function through degranulation [10]. Another subset of pro-inflammatory cells comprises the CD4+ T helper 1 (Th1) T lymphocytes and B cells. These cells are characterized as pro-inflammatory due to their capacity to trigger the activation of CD8+ T cells through the secretion of IFNγ and IL-2 [11,12]. Consequently, IFNγ and IL-2 are categorized as Th1 cytokines. The anti-inflammatory cells that can be found in the TME include M2 macrophages, myeloid-derived suppressor cells (MDSCs), CD4+ T regulatory cells (Tregs) and CD4+ T helper 2 (Th2) lymphocytes. These cells possess the capability to release cytokines such as IL-10 and IL-4 [13,14]. Notably, these cytokines play a role in suppressing the Th1 immune response within the tumor milieu. Ultimately, cytokines, as potent immune modulators, can exert potent beneficial but also detrimental effects upon the natural antitumor response. For instance, cytokines typically classified as immune response-inhibitory cannot be categorically labeled as pro-tumoral cytokines, as exemplified by the case of IL-10, which has demonstrated the ability to induce both tumor rejection and foster long-lasting tumor immunity [14]. Conversely, TNFα, which was originally named based on its direct cytotoxic activity on tumor cells, has been shown to participate in both the initiation and progression of cancer [15,16,17]. Finally, transforming growth factor β (TGFβ) is another double-edged sword cytokine which can have anti-tumor activity via its direct growth inhibition activity on cancer cells but which can also exert anti-inflammatory and immunosuppressive functions contributing to the immunosurveillance escape and oncogenesis [18].

Beyond their role in the immune system, cytokines are also able to exert a direct influence on pre-malignant and cancerous cells. While certain cytokines, such as members of the IFN family, can directly induce apoptosis in tumor cells [19,20], some cytokines such as IL-6 [15,21], TNFα [22], and TGFβ [23] can enhance cell proliferation and resistance to induced cell death, ultimately fostering tumor growth and progression [24].

The understanding of cytokine functions on cancer and immune cells has led to an increasing interest in their use as drugs. The administration of recombinant IFNα was shown to trigger the phagocytosis of tumor cells by macrophages and extended the survival of mice in multiple xenograft tumor models [25], and recombinant IFNα was the first cytokine-based drug approved for clinical use in 1986 for hairy cell leukemia, followed by IFNγ in 1991 for chronic granulomatous disease [26]. This study showed the potential of recombinant cytokines as therapeutics for cancer and the approval of recombinant IL-2 (Proleukin) for metastatic renal cancer quickly followed [27].

Although the immunotherapeutic potential of cytokines has proven to be widely useful in various diseases including hepatitis B and hepatitis C, chronic granulomatous disease, multiple sclerosis, rheumatoid arthritis, psoriasis, Crohn’s disease, and ulcerative colitis [28], the success of cytokine therapies in cancer has been limited by their poor bioavailability, inadequate pharmacokinetics (PK), and toxicity profile at therapeutic doses [29]. Therefore, maintaining therapeutically active concentrations in patients without inducing significant side effects remains a current challenge in the field of cytokine therapy.

To address the poor bioavailability issues observed with cytokine treatment, numerous vectorization technologies have been developed. PEGylation, which aims to prolong the half-life of cytokines, has been one of the most frequent approaches to improve the pharmacological profile of cytokines. PEGylation is clinically approved for IFNα2a [30,31] and is currently under evaluation for IL-2 [32], IL-10 [33], and IL-15 [34,35]. However, PEGylation is often accompanied by an unwanted effect on bioactivity, particularly resulting from the amphiphilic nature of PEG, as exemplified by PEGylated IFNα2a that displays only 7 % in vitro activity of the native cytokine [36]. Other approaches to improve the pharmacological profile of therapeutic cytokines include fusing them to antibody Fc-domains [37,38,39], albumin [37,40], or collagen binding lumican [41] resulting in extended half-lives of cytokines. However, novel approaches are still critically needed to reduce the toxicity of cytokine treatment and to improve their efficacy.

Among the promising technologies, antibody–cytokine fusion is one of the most advanced vectorization techniques showing anti-cancer efficacy. This review is focused on antibody–cytokine fusion proteins, also named immunocytokines (ICKs), developed for cancer treatment with a focus on the most impactful cytokines IL-2, TNFα, IL-10, IL-12, IL-15, IL-21, IFNγ, GM-CSF, and IFNα, with a particular emphasis on their immunomodulatory effects associated with their technological design. An overview of the pharmacological effects of the naked cytokines and ICKs tested for cancer therapy is detailed. Finally, some perspectives on this exciting field of biopharmaceuticals are discussed.

**Figure 1 pharmaceutics-16-00974-f001:**
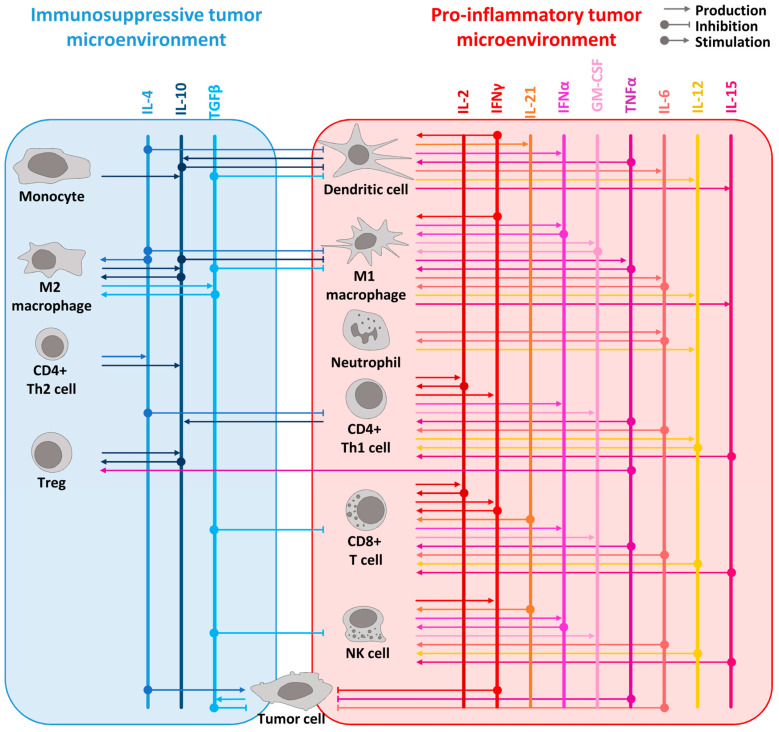
Overview of the cytokine network in pro-inflammatory and immunosuppressive tumor microenvironment. The principal immunosuppressive cytokines are listed in blue shades at the top of the left panel whereas the main pro-inflammatory cytokines are in red shades at the top of the right panel. The cytokines are represented as vertical lines. Arrows from cells to the cytokines represent the production of a cytokine. Arrows from the cytokines to the cells represent its stimulatory or inhibitory effect on the designated cells. For example, pro-inflammatory cytokines such as IL-2, IL-12, IL-15, IL-21, IFNα, and TNFα are known to stimulate lymphocyte proliferation and activation, while IFNγ can activate both lymphocytes and myeloid cells. IL-15 is produced by dendritic cells and macrophages, like IL-12, which is also produced by neutrophils. IFNα is produced by pro-inflammatory immune cells such as lymphocytes, macrophages, and dendritic cells. TNFα, mainly produced by macrophages, acts as a proinflammatory cytokine through its receptor TNFR1 but leads to anti-inflammatory effects by stimulating Tregs through its receptor TNFR2. GM-CSF, which is produced by macrophages, T cells, and NK cells, elicits an anti-tumor immune response by specifically stimulating myeloid cells and, notably, macrophages. Additionally, IL-10 promotes an anti-tumor response by inducing the expression of IFNγ and enhancing CD8+ T cell activity. To simplify the scheme, direct effects of cytokines on tumor cells are not represented. However, many cytokines have a direct effect on cancer cells such as TGFβ which acts as an anti-tumoral factor in the early stages of tumorigenesis [42] or IL-4 with its pro-tumoral effect [43].

## 2. Methods

PubMed was used to find published studies. The search was focused on selected cytokines known to exert an important role in cancer: IL-2, IL-15, TNFα, IL-10, IL-12, IL-21, GM-CSF, IFNγ, and IFNα. The two following key word associations were used:

(1) (Immunocytokine [Title/Abstract]) AND (cancer [Title/Abstract]);

(2) (“cancer” [Title/Abstract] AND “cytokine name” [Title] AND (“immune cytokine” [Title/Abstract] OR “immunocytokine” [Title/Abstract] OR “fusion protein” [Title/Abstract] OR “antibody cytokine fusion” [Title/Abstract] OR “antibody fusion” [Title/Abstract])).

The obtained results were then filtered to focus on ICKs designed to target the cytokines in the TME and tested in vivo.

Clinicaltrials.gov was used to find all clinical trials involving ICKs. The following keywords were used to search for trials: “cytokine name” fusion protein, “cytokine name” immune cytokine, “cytokine name” monoclonal antibody–cytokine fusion, “cytokine name” antibody fusion, “cytokine name” antibody–cytokine fusion. Among the search results, only ICKs for the treatment of cancer were selected.

A list of abbreviations is given in Appendix A.

## 3. Immunocytokines

ICKs are biotherapeutics using the advantageous properties of immunoglobulins by genetically fusing full antibody or antibody fragment coding sequences with a cytokine gene to limit the side effects and improve the poor PK observed with naked cytokine administration. They can be categorized into two types. The first type consists of large fusion proteins with cytokines fused to an intact immunoglobulin (Ig), while the second type comprises smaller fusion proteins based on antibody fragments such as single-chain variable fragment (scFv), F(ab’), F(ab’)2, tandem diabody (also called single-chain diabody), and tribody [44] (Figure 2). ICKs under clinical development are mostly fused to intact IgG and scFv but emerging diabodies [45] and tribodies [46,47] ICKs have been tested in preclinical models. The smaller fusion proteins display significantly reduced molecular masses, typically below 100 kDa in contrast to the Ig fusions, which have molecular weights in the range of 175 kDa [48]. Both the antibody format and cytokine composition play a significant role in determining the size, efficacy, and PK/pharmacodynamic (PD) profile of the resulting ICKs [49,50]. While Ig-based ICKs have several strengths such as a long serum half-life or high target avidity, their elevated size is associated with low tumor penetration resulting in low target exposure. Conversely, low-molecular-weight ICKs benefit from high tumor penetration but a shorter half-life.

The longer half-life observed with Ig-fused ICKs is mostly explained by the fragment crystallizable (Fc) region of the antibody used, which is responsible for the non-specific binding to Fc receptors (FcRs) expressed on immune cells [51]. The catabolic elimination rate of Ig-fused ICKs depends on the Ig class, which is largely influenced by the binding to a specific FcR, the neonatal FcR (FcRn). IgG1, IgG2, and IgG4 isotypes have very long half-lives of 18–21 days whereas IgG3, IgM, and IgA have shorter half-lives of 5–8 days [1]. The non-specific binding of the Fc region to FcγRI, FcγRII, and FcγRIII is responsible for IgG effector functions such as antibody-dependent cellular cytotoxicity (ADCC) or complement-dependent cytotoxicity (CDC) involving the immune cells. On the one hand, ADCC and CDC can increase the antitumor efficacy of an ICK, but on the other hand, they can also trigger off-target toxicity which can constitute a significant weakness of IgG-fused ICKs. Beyond the advantage of being able to modulate the cytokines’ PK profile and to engage immune cells through effector functions, ICKs offer the possibility to vectorize a payload toward the tumor site by targeting tumor-associated antigens (TAAs) or tumor-specific antigens (TSAs) through their variable region. The affinity of the ICKs for the TAAs also impacts the PK through target-mediated drug disposition. The differential expression of the TAAs and TSAs in healthy versus tumor tissues is a critical parameter to ensure a higher accumulation of the ICKs at the tumor site and limit the cytokine-mediated toxicity in healthy organs.

**Figure 2 pharmaceutics-16-00974-f002:**
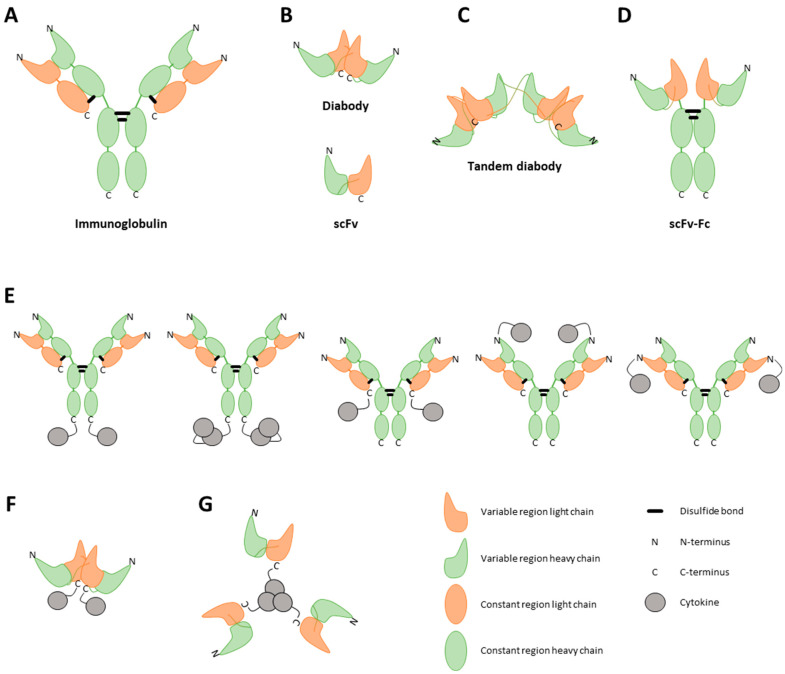
Visual representation of different immunocytokine (ICK) formats. ICKs can consist of a full immunoglobulin (**A**), a single-chain variable fragment (scFv) (**B**, bottom), a diabody (**B**, top), or a tandem diabody, also called a single-chain diabody (**C**). An scFv consists of the variable regions of a heavy and light chain. A diabody consists of two connected scFvs. To enhance the half-life of the construct, Fc regions can be strategically linked to antibody fragments (**D**). Furthermore, cytokines can be incorporated into the fusion protein as either monomers, dimers, or trimers, tethered to either the C-terminal or N-terminal regions of both the heavy and light chains of the antibody (**E**) or of the diabody (**F**) [52]. The choice of cytokine fused to the antibody profoundly influences the final structural conformation of the fusion protein. As an illustrative example, TNF, known to naturally assemble into trimers, when combined with an scFv, results in a final fusion protein adopting a trimeric configuration (**G**) [47].

The ICKs described hereafter have been developed using different cytokines, various technologies, and several TAAs have been utilized to deliver the cytokines to the TME (Appendix A). The antibodies used for ICKs are chosen based on the cancer indication of interest and the TAAs/TSAs which were found differentially expressed in the different given tumor type. While some TAAs/TSAs do not have any intrinsic function and are only used to vectorize the ICKs to the TME, some other antigens can exert direct or indirect tumorigenic functions. They enable to synergize the therapeutic potential of the blocking antibody with the immunostimulatory effects of the cytokine delivery in the TME, which constitutes a critical advantage of the newly developed ICKs.

## 4. IL-2

IL-2 is a pivotal factor in the growth, survival, and differentiation processes of T lymphocytes and remains one of the most extensively investigated cytokines [53,54,55,56]. IL-2 has a molecular size of 15 kDa and assumes a structural conformation comprising four α helices [53]. Functionally, IL-2 operates as a monomer, primarily being released from CD4+ T cells. Originally, IL-2 was called T cell growth factor [1] as it stimulates proliferation of antigen-stimulated T cells and NK cells by binding to the β and γ receptor chain of the IL-2 receptor (IL-2R) [57]. IL-2 is also able to stimulate Tregs by binding to the IL-2Rα chain [53]. Cancer cells from solid tumors lack the expression of IL-2R, signifying that the therapeutic impact of administered IL-2 can be primarily attributed to its immunological effects that promote the cytolytic activity of both NK and CD8+ cell types [58].

Recombinant IL-2, Proleukin, was approved for clinical use in 1985 for the treatment of metastatic renal carcinoma and melanoma [59,60,61]. Proleukin experiences rapid clearance resulting in a short half-life (±7 min) and achieving therapeutically active concentrations requires high doses or extended administration at low doses. High doses are frequently accompanied with severe side effects such as fever, malaise, arthralgias, hypotension, capillary leak syndrome, and severe influenza-like symptoms [62]. Low doses are accompanied by extended administration and long hospitalization periods. The clinical use of recombinant IL-2 is therefore in decline due to its poor PK and toxicity [63], so ICKs have been tested.

### 4.1. IL-2 Ig-Fused ICKs

The first ICK ever to be investigated in clinical trials to overcome recombinant IL-2-related issues is the hu14.18–IL-2 fusion protein [64,65,66]. The fusion consists of two IL-2 molecules linked to the C-terminal domain of the heavy chain of the humanized 14.18 IgG1 monoclonal antibody (mAb) that recognizes disialoganglioside (GD2) expressed on neuroblastoma and melanoma cells. Hu14.18–IL-2 is the humanized version of ch14.18–IL-2, that showed antitumor activity in murine models [66,67,68]. The fusion of IL-2 to 14.18 was supported by a clinical trial demonstrating the benefit of combining free recombinant IL-2 with the anti-GD2 14.G2a and was rationalized mechanistically by the stimulatory effect of IL-2 on NK cells and the resulting ADCC-mediated effect through the anti-GD2 mAb [69]. Four phase II clinical trials have been performed to assess the activity of the construct against melanoma and neuroblastoma [70,71,72,73]. Around 20% of the neuroblastoma patients responded to the treatment. In melanoma, hu14.18–IL-2 induced immune activation with some antitumor activity [74], particularly in patients with pre-existing tumor immune gene signatures [75].

The cytokine fusion to the C-terminal domain of the Ig heavy chain can impact the FcR and FcRn binding of the ICK that may decrease its effector function and half-life. Interestingly, IL-2 fusion to the light chain or N-terminal domain of the heavy chain is an option to overcome this issue. IL-2 fusion to the light chain C-terminal of the anti-GD2 14.18 antibody has been shown to improve the ICK half-life following subcutaneous administration and improved effector activities of ADCC and CDC in animal models [76].

In B cell lymphoma, promising data were obtained in both preclinical models and in the clinic with administration of IL-2 plus rituximab (anti-CD20) combination [77,78,79]. Based on these results, a B cell-targeting ICK was developed by fusing IL-2 to an anti-CD20 (DI-Leu16-IL-2) IgG antibody for this indication [79]. Interestingly, DI-Leu16-IL-2 was more effective than combinations of anti-CD20 antibodies and IL-2 in the preclinical model [79,80,81]. In addition, these studies concluded that the therapeutic effect of DI-Leu16-IL-2 was mainly due to the delivery of active IL-2 rather than mediated by ADCC. Phase I/II studies, conducted on patients with B cell lymphoma, showed lymphocyte expansion [81], and partial responses were observed in five out of 13 patients, while three out of 13 patients exhibited complete responses [80] when treated with DI-Leu16-IL-2.

Tucotuzumab celmoleukin (EMD 273066/huKS–IL-2) consists of two IL-2 molecules fused to the Fc part of a humanized IgG mAb-targeting epithelial cell adhesion molecule (EpCAM or KSA). EpCAM is highly expressed on many epithelial cancer types among which are prostate, colon, breast, and lung cancer [64,82]. Two phase I trials have been performed using huKS–IL-2 in several solid tumors [83,84,85,86]. Most patients showed a transient immune response to huKS–IL-2 and 38% of all patients showed a stable disease as best response [85]. In subjects with extensive-disease small cell lung cancer, huKS–IL-2 in combination with cyclophosphamide was well tolerated but did not show progression free survival (PFS) or overall survival (OS) benefits compared with best supportive care alone in a phase II study [87].

A different IL-2-based construct was designed to target phosphatidylserine (PS). PS is a type of phospholipid externalized on the outside of tumor cell and tumor endothelial cell membranes. Exposed PS induces the production of multiple mediators in the TME. Furthermore, 2aG4–IL-2 is an anti-PS antibody (2aG4) fused to IL-2 which takes advantage of IL-2 immunostimulatory activities and the several modes of action of the 2aG4 antibody. This antibody demonstrated tumor growth inhibition efficacy as a single agent in multiple tumor models by destroying tumor vessels through ADCC against the tumor vascular endothelium and by blocking the immunosuppressive signals from PS [88]. Mice treated with 2aG4–IL-2 had a longer survival rate compared to the non-tumor targeted IL-2 control group in the context of vaccination [89].

Carcinoembryonic antigen (CEA) is a highly expressed TSA in the majority of colon cancers, as well as many other solid tumors such as ovarian, breast, and pancreatic tumors [90,91,92,93]. M5A–IL-2 is an anti-CEA human IgG1 antibody fused to IL-2. Even though M5A–IL-2 did not improve the antitumor efficacy compared to IL-2–Fc fusion protein, higher tumor uptake and slower blood clearance were observed in M5A–IL-2-treated mice [94]. Of note, the fusion of IL-2 to Fc-derived immunoglobulins with high effector function (such as mouse IgG2 or human IgG1) showed constructs with low toxicity and high antitumor activity supported by Fc-mediated Treg depletion and moderate systemic immune cell activation [95,96]. This demonstrates the importance of selecting appropriate IgG subtypes for ICK design.

Prostate-specific membrane antigen (PSMA) is expressed in all forms of prostate tissue, including carcinomas, and is an attractive target for treatment of prostate cancer [97]. KM2812 is a C-terminus IL-2-fused anti-PSMA mouse–human chimeric IgG1 ICK. KM2812 was only tested in a xenograft model in which it exhibited antitumor activity [98].

Human epidermal growth factor receptor 2 (HER2/neu) is a well-recognized TAA in breast and ovarian cancers which are treated with the anti-HER2 mAb trastuzumab. Anti-HER2/neu IgG3–(IL-2) has been tested in preclinical models with the aim to improve the partial success of trastuzumab observed in the clinic. This fusion protein demonstrated promising efficacy associated with an increased antigen presentation efficacy in mouse models [99] but no further development was reported, possibly because of the limitation of using wild-type IL-2 described hereafter.

Although the above-mentioned constructs enabled to drive IL-2 targeting to the TME, some adverse effects of IL-2 were still observed with some ICKs such as hu14.18–IL-2 or DI-Leu16-IL-2. The characterization of the IL-2 signaling pathway enabled to better understand the underlying molecular mechanisms responsible for IL-2-related toxicity and led to the development of several engineered IL-2 cytokines which have been used for ICK design.

### 4.2. Engineered IL-2 Ig-Fused ICKs

Hu14.18–IL-2 has been associated with fever, chills, fatigue, pruritus, and myalgias, and severe toxicities, especially those of the vascular compartment such as capillary leak syndrome [100,101,102]. The activation of cells expressing the intermediate-affinity IL-2 receptor in the vascular compartment was proposed to be responsible for the vascular toxicity by inducing inflammatory cytokine release by NK and other cells in response to hu14.18–IL-2 [103,104]. To overcome this issue, IL-2 has been engineered with D20T mutation (IL-2LT) and fused to the C-terminal part of the heavy chain of an NHS76 IgG2 antibody, recognizing DNA structures in the necrotic core of a tumor [105]. The IL-2LT molecule demonstrated a reduction in weight loss and an extended time interval before the onset of toxic symptoms in mice, as compared to recombinant IL-2 [105,106]. The resulting construct selectikine (NHS-IL-2LT) was assessed for patients with metastatic, locally advanced solid tumors, and B cell lymphoma in two phase I clinical trials and a phase II clinical trial which was terminated due to discontinuation of the program by the leading pharmaceutical firm [107,108,109]. Although the efficacy was left unknown, NHS-IL-2LT induced strong activation of both CD4+ and CD8+ T cells and only weak NK cell activation in line with a reduced activation of the intermediate-affinity IL-2 receptor on NK cells [106].

Another undesirable activity of IL-2-based ICKs is the induction of Treg expansion through the binding of IL-2 to IL-2Rα observed in patients dosed with NHS-IL-2LT [106] and DI-Leu16-IL-2 [81]. Since Tregs are well known for their immunosuppressive activity favoring tumor progression, development efforts have been made to engineer a recombinant IL-2 with low stimulating activity on Tregs. This led to the discovery of IL-2 variant (IL-2v) harboring a mutation enabling the preferential activation of T cells through IL-2Rβγ heterodimeric receptor without activating IL-2Rα (CD25) and, consequentially, not stimulating Tregs [110,111,112,113,114,115]. IL-2v has been used to generate ICKs targeting several antigens which all have reached the clinical phases: CEA (CEA-IL-2v, RG7813, cergutuzumab amunaleukin), fibroblast activation protein α (FAP) (FAP-IL-2v, RO6874281/RG7461, simlukafusp alfa), and PD-1 (RO7284755, RG6279, eciskafusp alfa).

CEA-IL-2v is composed of IL-2v fused to the C-terminus of a CEA-specific antibody devoid of Fc-mediated effector functions [116]. Consistently with the reduced binding of IL-2v to CD25, CEA-IL-2v induced a CD8+:CD4+ ratio toward CD8+ T cells both in the periphery and in the tumor associated with single-agent and combination efficacy in syngeneic models.

FAP-IL-2v is an IL-2v cytokine fused to the human IgG1 antibody 4B9 with abolished FcγR binding but functional FcRn binding to retain the advantage of IgG-like antibody half-life [117]. FAP-IL-2v demonstrated improved tumor targeting compared to FAP-IL-2wt and showed antitumor efficacy in combination with anti-PD-L1 immune checkpoint blocker [118].

PD-1-IL-2v is an anti-programmed cell death 1 (PD-1) antibody fused to IL-2v via its C-terminal. PD-1-IL-2v differentiates from CEA-IL-2v and FAP-IL-2v ICK from a targeting point of view since PD-1 is expressed on immune cells, particularly T lymphocytes, and has been well characterized as an immune checkpoint involved in T cell exhaustion and immunosuppression. PD-1 binds PD-L1 expressed on tumor cells or myeloid suppressive immune cells, resulting ultimately in tumor immunosurveillance escape [119]. The discovery of this mechanism, that also involves other immune checkpoint proteins, revolutionized the therapy of cancer with the development of immune checkpoint blockers such as anti-PD-1 antibodies. Therefore, the primary objective supporting the PD-1-IL-2v design was to obtain a construct with high immunostimulatory properties by taking advantage of both PD-1 blockade and IL-2 functions. RO7284755 treatment in combination with anti-PD-L1 elicited efficacious antitumor immunity by promoting pre-existing stem-like and tumor-reactive CD8+ T cells and remodeling immunosuppressive tumor-associated macrophages [115]. These data led to evaluation of PD-1-IL-2v in a phase I clinical trial to assess its safety profile across various dosage levels and to compare its efficacy both as a standalone treatment and in combination with anti-PD-L1 therapy, for the management of advanced and metastatic solid tumors.

IBI363, another anti-PD-1 antibody fused to a modified IL-2 (IL-2m) engineered to restore some CD25-mediated antitumor activities with a better safety profile than IL-2v [120], is also in clinical development.

SumIL-2 is another example of IL-2 engineering strategy designed to decrease CD25 binding while preferentially increasing IL-2Rβ binding to allow more-efficient cytotoxic T lymphocyte targeting in the TME. SumIL-2 has been fused to cetuximab (Erb-sumIL-2) to target epithelial growth factor receptor (EGFR) or to trastuzumab (a-HER2-sumIL-2) to target HER2/neu. Erb-sumIL-2 featuring an anti-human EGFR in one arm and a sumIL-2-Fc monomer in the second arm showed strong antitumor efficacy dependent on the induction of tumor-infiltrating CD8+ lymphocytes in animal models [110]. In the same study, the a-HER2-sumlL2 showed efficacy in combination with the tyrosine kinase inhibitor afatinib, although its effect on the antitumor immune functions was not documented.

### 4.3. Other IL-2 Fusion Proteins

Non-Ig ICK formats have been developed in order to take advantage of lower-molecular-weight proteins to facilitate the ICK penetration in the TME. Darleukin (L19-IL-2) and Teleukin (F16-IL-2) are two scFv diabody type IL-2 fusion ICK-targeting extracellular matrix components [50] that are still being clinically assessed. L19-IL-2 is composed of an antibody targeting the extracellular domain B (EDB) of fibronectin fused to two IL-2 molecules on the C-terminal domain of the heavy chains. L19-IL-2 has been assessed in clinical trials for the treatment of advanced solid tumors as monotherapy or in combination with existing chemotherapies. A two-armed phase II clinical trial in 69 patients suffering from stage IV melanoma concluded a significant increase in progression-free survival for patients receiving the L19-IL-2 plus dacarbazine combination therapy in comparison to chemotherapy alone [121]. F16-IL-2 uses another antibody (F16) which targets the Tenascin-C glycoprotein found in the extracellular matrix. F16-IL-2 showed beneficial signs and acceptable tolerability in combination with chemotherapy for the treatment of solid tumors and metastatic breast cancer [122]. F16-IL-2 has also shown promising anti-cancer effects and safety in patients suffering from acute myelogenous leukemia relapsed after allogeneic hematopoietic stem cell transplantation [123]. The most recent clinical trial investigates F16-IL-2 in combination with anti-PD-1 or chemotherapy.

Among the tumor antigens, some cell-surface proteins such as p53 or melanoma-associated antigen recognized by T cells (MART-1) are hardly druggable by classical antibodies because they require recognition in the context of major histocompatibility complex (MHC) class I presentation. T cell receptor (TCR) or antibody-like TCR have been designed to overcome this limitation. However, only few TCR-based ICKs have been reported, most likely because TCR exhibits overall reduced affinity of antigen binding and has slower on-rate binding which might not favor ICK accumulation in the TME compared to Ig-fused ICKs.

P53 tumor suppressor protein is overexpressed in a broad range of human malignancies. In addition, 264scTCR/IL-2 is a soluble single-chain TCR fusion protein constituted of IL-2 and a three-domain HLA-A2-restricted TCR specific for a peptide epitope of the human p53. Moreover, 264scTCR/IL-2 promoted NK cell infiltration into tumors [124] and reduced lung metastases in a mouse model [125].

MART-1scTCR-IL-2 is another example of a TCR-based ICK that has a similar design than 264scTCR/IL-2 but recognizes MART-1 antigen which is found on the surface of melanocytes and can be used to diagnose melanoma. MART-1scTCR-IL-2 has not been tested in a MART-1-positive tumor model but showed partial antitumor efficacy in a lung cancer model tumor in which the increased half-life of IL-2 was propose as a supporting factor underlying efficacy [124].

Finally, GI-101 is a non-Ig ICK which consists of the extracellular domain of CD80 acting as a cytotoxic T lymphocyte-associated protein 4 (CTLA-4) inhibitor linked to IL-2v through an IgG4 Fc fragment. CTLA-4 is constitutively expressed by Tregs but only upregulated in conventional T cells after activation. It binds to CD80 or CD86 expressed on the surface of antigen-presenting cells, a phenomenon which is particularly observed in cancers [126]. GI-101 inhibited tumor growth and induced a robust increase in M1 macrophages and CD8+ central memory T and NK cells, but not Tregs, in the TME of a syngeneic tumor model [127].

## 5. IL-15

IL-15 is a 13 kDa protein that exhibits effects similar to those of IL-2 [1]. This resemblance is attributed to the fact that IL-15 utilizes the same β and γ IL-2R chains for signaling as IL-2 but does not activate the IL-2Rα on Tregs [128]. IL-15 also diverges in its exclusive receptor, IL-15Rα [128]. Despite its comparable impact, IL-15 has garnered notably less attention in the field of ICKs compared to IL-2. IL-15 functions as a heterodimer cytokine complexed with IL-15Rα for stability [128] and promotes the growth and functions of NK cells and cytotoxic CD8+ T cells [129,130]. The antitumor effect of the cytokine directly depends on its ability to activate these two cell types. Compared to IL-2, toxicity of IL-15 is lower due to its ability to stimulate cytotoxic T cells without stimulating Tregs [131]. However, high dose recombinant IL-15 is associated to reduced appetite, diarrhea, and weight loss, without autoimmune manifestations or infections [132]. In addition, the half-life of IL-15 is around 1 h, making it hard to meet a therapeutic window [133]. Use of a truncated IL-15Rα composed of the sushi domain of the receptor has been successfully employed. Also called receptor-linker-IL-15 (RLI) (SO-C101), the fusion molecule of human IL-15 covalently linked to the human IL-15Rα sushi+ domain was employed for retaining IL-15 stability and binding capabilities [134].

### 5.1. IL-15 Ig-Fused ICKs

The anti-GD2-RLI and hu14.18–IL-15 are two ICKs designed to target IL-15 to GD2-positive tumors. The anti-GD2-RLI consists of an IgG1 anti-GD2 antibody fused to the N-terminal domain of human IL-15Rα linked via a 20-amino-acid sequence to human IL-15. This construct showed antibody effector functions and displayed strong antitumor activities in syngeneic cancer models. Interestingly, the anti-GD2-RLI had a higher therapeutic potency than those of IL-15Rα/IL-15 and anti-GD2 alone or in combination [135]. Hu14.18–IL-15 has a similar design to hu14.18–IL-2 described above but demonstrated higher efficacy in syngeneic mouse models with orthotopic neuroblastoma despite showing equivalent ADCC [136].

Combination therapy with recombinant IL-15 and blocking antibodies against CTLA-4 has been shown to increase survival and reduce tumor growth in murine tumor models, as compared to one of the treatments alone [137]. This led to the development of JK08 which is an ICK targeting the immune checkpoint family on T cells through a CTLA-4-targeting antibody connected to an IL-15/sushi domain fusion peptide. JK08 can elicit ADCC-mediated killing of CTLA-4-expressing cells such as Tregs enhancing the antitumor responses through depletion of Tregs and stimulation of NK and CD8+ T cells through IL-15 signaling [138]. JK08 is being evaluated in a phase I clinical trial.

KD033 and its murine cross-reactive surrogate srKD033 are anti-PD-L1/IL-15 fusion proteins engineered to have minimal effector functions via the introduction of the LALA mutation in the Fc portion which minimize the ADCC and CDC. IL-15 was linked to the C-terminal Fc domain with the intention to likely preserve high-affinity binding to PD-L1, favored by the opposite sites of binding, on either tumor or immune cells of the TME. Consistent with this mode of action, srKD033 demonstrated high retention in the TME and showed efficacy in several cancer models, including tumors known to be non-responsive to immune checkpoint blockers, such as the B16-F10 melanoma model [139].

GT-00AxIL-15 is an IL-15-based ICK targeting the tumor-specific, glycosylated epitope of MUC1 (TA-MUC1). TA-MUC1 is a carbohydrate–protein epitope expressed on a variety of tumor types but which is absent on normal cells. The targeting moiety of GT-00AxIL-15 is the humanized IgG1 mAb GT-00A (gatipotuzumab) clinically developed for treatment of solid tumors [140]. GT-00AxIL-15 is constituted of two human wild-type IL-15 molecules linked to the Fc domain of GT-00A which are not precomplexed with the IL-15Rα sushi domain. This construct exhibited a favorable tumor accumulation with a 13-day half-life in mouse serum and mediated antitumor effects associated with robust immune activation and expansion in a mouse model [141].

### 5.2. Engineered IL-15-Fused ICKs

IL-15N72D is an IL-15 mutein engineered to exhibit superagonist activity through improved binding ability to the human IL-15Rβ chain. N-803/ALT803 is an IL-15N72D associated to IL-15Rα sushi domain fused to a fully human IgG1 fragment [142]. 2B8T2M ICK was generated comprising four single-chain anti-human CD20 Fv domain of rituximab linked to the N-termini of the IL-15N72D and IL-15RαSu-Fc of the N-803 fusion protein [143]. With its IgG1 domain, 2B8T2M displays a tri-specific binding activity to CD20, IL-15Rβγc, and FcγR [143]. Treatment of tumor-bearing mice with 2B8T2M was more effective than rituximab [144].

The N-803 mutein was also fused to an anti-PD-L1 double scFv fusion protein to form an ICK named N-809. N-809 had improved efficacy versus N-803 + anti-PD-L1 combination and induced enhanced CD8+ T and NK cell tumor infiltration, activation, and function in the tumor model [145].

Another IL-15 mutein was engineered through the insertion of targeted mutations (N1G-D30N-E46G-V49R-E64Q) to eliminate IL-15Rα binding and reduce IL-2Rβ/γ affinity. This IL-15 mutein was linked monovalently to the C-terminus of an anti-PD-1 IgG1 having reduced FcγR binding through a flexible “GGGGSGGGGSGGGG” linker. The anti-PD-1-IL-15m (PF-07209960) was designed to deliver PD-1-mediated and avidity-driven IL-2/15 receptor stimulation to PD-1 positive tumor-infiltrating lymphocytes while minimally affecting circulating peripheral NK and T cells. This construct is currently being assessed in clinical trials after having shown the induction of exhausted CD8+ T cell expansion in tumors and higher efficacy, without aggravation of body weight loss, compared to single-agent treatments with an IL-15 superagonist, an anti-PD-1, or the combination thereof in a mouse model [146].

### 5.3. Other IL-15 Fusion Proteins

Lymphocyte-activation gene 3 (LAG-3) is a cell surface molecule expressed on activated T and NK cells. Similarly to PD-1 and CTLA-4, LAG-3 can be considered as a TAA involved in the negative regulation of the immune system. Anti-LAG-3 antibodies have been developed on the same rationale as other immune checkpoint blockers [147]. An anti-LAG-3-IL-15 ICK has been designed aiming to synergize the immunostimulatory effects of both the LAG-3 blockade and IL-15. This construct contains a single-chain IL-15/IL-15Rα and LAG-3-targeting arms attached to a heterodimeric Fc region. Interestingly, treatment with the anti-LAG-3-IL-15, combined productively with anti-PD-1, promoted T cell expansion and showed minimal peripheral activity [148].

PFC-1 (BJ-001) is a fusion protein targeting tumor cells that overexpress integrins such as alpha v beta 3 (avb3), alpha v beta 5 (avb5), and alpha v beta 6 (avb6). BJ-001 is composed of an integrin-targeting fusion protein, an Fc domain (IgG1), and an IL-15/IL-15Rα complex. BJ-001 was brought to clinical trial in 2020 after demonstrating impressive antitumor and immunostimulatory activities in mouse models [39,149].

Several IL-15-fused ICKs targeting the antigen FAP mentioned above have been generated. First, scFv_RD_IL-15 is an ICK featuring IL-15, an extended IL-15Rα sushi domain, and an anti-FAP antibody scFv moiety. The construct demonstrated strong tumor growth inhibitory effect in the B16 metastasis model compared to untargeted IL-15 or compared to the construct without the sushi domain [134]. Based on this construct, trifunctional antibody-fusion proteins have been tested by incorporating an additional arm with binding affinity for costimulatory members of the TNFR superfamily expressed on T and NK cells, CD137 (4-1BB), glucocorticoid-induced TNF (GITR), or CD134 (OX40), to functionally stimulate the immune response in the TME. These constructs have demonstrated promising potential to improve T cell activation and antitumor activities [150,151].

Similarly to Darleukin, a first IL-15 fusion protein with the L19 antibody in diabody format was designed but showed a suboptimal biodistribution profile compared to previously tested L19 ICKs [152]. Novel ICKs termed F8-F8-IL-15 and F8-F8-SD-IL-15, recognizing the alternatively spliced extra-domain A (EDA) of fibronectin, rather than its extra-domain B (EDB) like the L19 antibody fragment, have been developed with IL-15. F8-F8-IL-15 and F8-F8-SD-IL-15 are similar scFv diabodies except that F8-F8-SD-IL-15 is constituted of the IL-15Rα sushi domain in addition of IL-15. Despite a strong preferential uptake in solid tumors and their capacity to increase the CD8+/CD4+ T cell ratio in the TME, these two constructs did not show strong antitumor efficacy as monotherapies in primary tumors [153]. However, in the same study, both products were active in combination with targeted TNF therapy in mouse models and the F8-F8-SD-IL-15 was more efficient in reducing tumor metastasis formation in the lung thanks to the presence of the sushi domain.

The activating receptor natural killer group 2, member D (NKG2D) is an activating receptor belonging to the NKG2 family of C-type lectin-like receptors which is mostly expressed by NK cells and controls both adaptive and innate immunities [154]. Interaction of NKGD2 with its ligands (NKGD2L) leads to NK cell-mediated cytotoxicity. However, it was demonstrated that high serum concentrations of soluble NKG2DLs may suppress tumor immunity and NK cell activity via downregulation, contributing to an escape from the tumor immunosurveillance machinery [155,156]. DsNKG2D-IL-15 is an ICK comprising double NKG2D extracellular domains fused to IL-15 designed to target and stimulate NK cell activities. DsNKG2D-IL-15 administration increased the frequencies of NK but also CD8+ T cells in the TME and retarded the tumor growth in colon cancer models [157].

HCW9218 is a heterodimeric bifunctional fusion molecule consisting of the extracellular domains of the human TGFβ receptor II and the human interleukin-15 (IL-15)/IL-15 receptor α complex [158]. TGFβ exerts tumor-promoting effects via several mechanisms including immune suppression. Treatment with HCW9218 could block TGFβ functions through sequestration and lead to the reduction of the immunosuppressive TME, enhancing immune cell infiltration and tumor growth inhibition in mouse models [158,159]. HCW9218 is currently being tested in a phase Ib/II clinical trial.

### 5.4. Attenuated IL-15 with Tumor Cleavable Masking Systems

Similarly to N-809 or KD033, LH05 is also an anti-PD-L1 ICK fused to IL-15. LH05 differentiates from N-809 mainly based on the structure and the linker which were designed to attenuate IL-15 activity when fused to the ICK, and to release the active IL-15/IL-15Rα sushi domain in a proteolytic cleavage-dependent manner in the TME. This was achieved by incorporating a protease-cleavable linker between the antibody and the IL-15/IL-15Rα sushi domain which can mask IL-15 activity by steric hindrance caused by the Fc fragment and the sushi domain. In that sense, LH05 can be consider as an anti-PD-L1/IL-15 prodrug in which IL-15 is attenuated until reaching the TME, thereby preventing the “cytokine sink” effect and systemic IL-15 activities. The antitumor efficacy of LH05 was shown in an animal model and was associated with NK and CD8+ T cells recruitment/activation in cold tumors and demonstrated reduced systemic toxicity when compared to non-cleavable anti-PD-L1/IL-15 [160].

## 6. TNF

Tumor necrosis factor α (TNFα) was first identified in the 1970s, and a diverse range of functions were revealed [161,162] following the cloning of the gene in 1984. TNFα is predominantly synthesized in its full, membrane-bound form, organized as homotrimers [163]. The biologically active soluble form of TNF, measuring 17 kDa, is liberated through proteolytic cleavage by the metalloprotease TNFα-converting enzyme [164]. TNF’s role in tumor immunology is akin to a double-edged sword. On the one hand, systemic administration of recombinant TNFα (Beromun) has been approved for the treatment of patients suffering from soft-tissue sarcoma. TNFα inhibits tumor growth through several mechanisms. Firstly, the protein is directly cytotoxic for tumor cells. Secondly, TNFα promotes the expansion of activated B and T lymphocytes and facilitates the generation of cytotoxic T cells. In addition, it triggers the activation of monocytes/macrophages and granulocytes, resulting in heightened phagocytic activity, respiratory burst, degranulation, and enhanced adherence to endothelial cells [165]. However, this cytokine treatment is also limited in its therapeutic use due to its substantial side effects, like shock and organ failure, and short half-life [166,167]. On the other hand, in cancer resistant to TNFα-induced cytotoxicity, TNFα has the capacity to incite proliferation, enhance cancer cell survival, promote migration, and stimulate angiogenesis, resulting in tumor promotion [165].

Apart from an ICK named ZME-TNF, which binds to the glycoprotein 240 antigen and was the only TNF-IgG fused construct tested [168,169] but poorly documented, the scFv format was mostly used to design TNF-ICK as it favors the formation of trimeric TNF like endogenous homotrimers [47]. In addition, the smaller scFv and diabodies are more suited to decrease systemic cytokine levels [170].

A clinically advanced construct carrying TNF is the L19-TNF construct (Fibromun) which was designed using the same scFv antibody as Darleukin. Initially, monotherapy showed promising results in preclinical studies using mouse models of fibrosarcoma and colorectal cancer [171]. Tumors showed inhibited growth when compared to untargeted TNF and most tumors became necrotic upon a single i.v. administration of L19-TNF. However, the monotreatment in patients suffering from advanced solid tumors in a phase I/II clinical trial did not lead to any objective response [172]. In contrast, a combination treatment with chemotherapy induced an objective response in 89% of the patients suffering from advanced extremity melanoma [173].

However, the most clinically advanced TNF ICK treatment is a combination therapy with L19-TNF and L19-IL-2 called Nidlegy. The L19-IL-2/L19-TNF combination showed a complete response of some melanoma lesions upon intralesional administration [174]. In addition, around 50% of the non-injected lesions showed a complete response suggesting a systemic response [174]. The most advanced clinical trial, the phase III NeoDREAM trial, assesses the effect of L19-IL-2 and L19-TNF combination as neoadjuvant therapy in stage IIIB/C melanoma patients. As of today, no results of the trial have been published yet. The most recent clinical trial, assessing this combination, started in April 2022. The phase II, open-label, multicentric, proof-of-principle basket trial is recruiting 70 patients suffering from malignant tumors of the skin amenable to intra-tumoral injection.

The above-mentioned F8 scFv, targeting fibronectin as well, was fused to murine TNF (F8-TNF) [175]. The construct has been tested as a mono-treatment, or in combination with chemotherapy, in murine models of sarcoma in pre-clinical trials [175,176]. When used as a mono-therapy, F8-TNF demonstrated significant inhibition of tumor growth compared to the control group and resulted in complete cures for 40% of the animals [175]. Moreover, when combined with doxorubicin, F8-TNF exhibited an even stronger antitumor effect and led to complete eradication of tumors [175]. Further experiments revealed that when mice were rechallenged again with WEHI-164, a fibrosarcoma model, or heterologous C51 or CT26 colon cancer models, tumors did not grow showing immune memory. Using a depletion strategy, the authors showed that this immune memory was mediated by CD8+ T lymphocytes [176].

Carbonic anhydrase IX (CA-IX) is a TAA induced by hypoxia, implicated in cancer invasiveness, and correlated with therapeutic resistance [177]. cG250-TNF is an ICK comprising IgG CH2/CH3 truncated chimeric G250 antibody directed against CA-IX and fused to TNF. cG250-TNF demonstrated antitumor activity which could be synergized by combination treatment of low doses of nontargeted IFNγ. Administration of cG250-TNF and IFNγ resulted in significant synergistic tumoricidal activity in a renal cancer model [178].

Human TNF was engineered to generate an attenuated TNF-ICK that could reduce systemic toxicity through the substitution of isoleucine for alanine (TNF^I97A^) and was fused to the L19 antibody fragment (L19-TNF^I97A^). L19-TNF^I97A^ showed decreased activity and binding on TNFR1 when compared to wild-type L19-TNF [171]. Interestingly, the biological activity of L19-TNF^I97A^ on TNFR1 was restored upon binding to the target antigen EDB. L19-TNF^I97A^ showed a potent antitumor effect without apparent toxicity compared with the wild-type protein.

A recombinant human TNFα (rmhTNFα) was fused to the GX1 peptide designed to target the gastric cancer vasculature [179,180]. The fusion protein that binds to Transglutaminase 2 (TGM2) was selectively delivered to targeted tumor sites, significantly improving the antitumor activity and decreasing the systemic toxicity of rmhTNFα.

## 7. IL-12

IL-12 is a cytokine consisting of heterodimeric subunits encoded by distinct genes: IL-12A (p35) and IL-12B (p40). The two separate proteins are linked through the formation of three disulfide bridges, resulting in the creation of the P70 fragment [181]. Functionally, this cytokine is classified as pro-inflammatory, as it elicits the differentiation of CD4+ Th1 T cells, promotes the development of cytotoxic CD8+ T cells, induces NK cells and NK T cells, and concurrently suppresses tumor-associated macrophages [182,183,184]. In addition, IL-12 induces the release of IFNγ [185] and has exhibited notable antitumor efficacy in numerous pre-clinical cancer models [186]. However, the efficacy of tolerable dose recombinant IL-12 in clinical trials showed no significant benefit [187,188,189,190]. As it is often the case with cytokines, the systemic administration of recombinant IL-12 is accompanied by pronounced adverse events [187,191]. Systemic IL-12 administration, in humans, is linked to a surge in systemic levels of IFNγ, TNFα, and IL-6 associated with a reduction in peripheral blood lymphocytes, monocytes, and neutrophils [187,191]. Since IL-12 can activate both the innate and the adaptive components of the immune system [186], it represents an ideal payload for ICK-based immunotherapies.

### 7.1. IL-12 Ig-Fused ICKs

Recently, IL-12 constructs have also been introduced to clinical trials. The most well-established IL-12 construct used in clinical trials is NHS-IL-12, which is composed of the previously mentioned NHS76 antibodies fused to the P70 heterodimeric form of IL-12 [192]. The construct showed an increase in host immunity with a reduction in immunosuppressive myeloid cells in the TME leading to a significant reduction in tumor growth in a murine bladder cancer model [193,194]. Since 2020, NHS-IL-12 entered in 10 clinical trials for the treatment of patients suffering from several types of cancers including breast, cervical, and pancreatic cancer. Early results show treatment-related adverse events including a decrease in circulating lymphocytes, an increase in transaminases, and flu-like symptoms, but most adverse events were low-grade [195]. In addition, an increase in frequencies of activated and mature NK cells and NK T cells was witnessed. Around 50% of the patients experienced a stable disease but no objective antitumor response was observed [195,196].

IL-12 has also been fused to another antibody, named BC1, targeting the extra-domain B (EDB) of fibronectin [197,198,199]. BC1-IL-12 (AS1409) showed antitumor activity in several human tumor models in SCID mice [200]. SCID mice lack functional T and B cells, suggesting that the therapeutic effect is NK cell-driven rather than B or T cell-driven. The construct was evaluated in patients with melanoma and renal cell carcinoma for its safety, efficacy, markers of IL-12-mediated immune response (IFNγ levels), and pharmacokinetics [197]. BC1-IL-12 was well tolerated and IFNγ levels were increased in all patients. In six out of 11 patients a stable disease was witnessed [197].

The chTNT-3/huIL-12 ICK was constructed by fusing IL-12 with the necrosis (single-stranded DNA)-targeting antibody chTNT-3. This fusion protein induced reduction in five prostatic tumors in a humanized mouse model [201].

### 7.2. Other IL-12-Fused ICKs

A novel antibody targeting FAP, named 7NP2, was recently fused to IL-12 in an scFv diabody format. This construct showed a potent antitumor activity in mouse models, both as a single agent and in combination with an immune checkpoint blocker. In non-human primates, the fully human construct was tolerated [202].

Another ICK carrying IL-12 is based around the L19 antibody, described above for IL-2 delivery, called IL-12-L19L19 (Dodekin). The construct is a product of a great extent of protein engineering for optimal IL-12 delivery [203,204,205,206,207]. IL-12 is fused to a single chain of the L19 diabody through its C-terminus by a 15-amino-acid linker [203]. The single chain still comprises two fused heavy chains and two light chains [208]. Linker optimization studies revealed that a 15-amino-acid linker (GSADGGSSAGGSDAG) displayed the best tumor-targeting properties [207]. Dodekin is currently being tested in a phase I clinical trial in patients with solid tumors.

The von Willebrand factor A3 domain has been shown to serve as a collagen-binding domain (CBD) enabling binding of fusion proteins to exposed collagen in disordered tumor vasculature [209]. A CBD-IL-12 ICK has been shown to remodel the TME of immunologically “cold” murine tumor models and to cooperate with immune checkpoint blockers to induce tumor eradication [190,210].

## 8. IL-21

IL-21 is a heterodimeric cytokine described for its pleiotropic effects [1]. In the context of tumor biology, its most pivotal role lies in the induction of cytotoxic T cells [211]. IL-21 alone does not initiate the development of CD8+ T cells, but it exhibits robust synergy with IL-7 or IL-15, bolstering CD8+ proliferation, enhancing functional responses, and promoting the production of IFNγ in mice [212]. Despite being evaluated in clinical trials, recombinant IL-21 has also encountered challenges related to toxicity and half-life [213]. This necessitates a critical focus on strategies such as half-life extension and localized delivery to optimize IL-21 antitumor efficacy.

Currently an IL-21 carrying human serum albumin (HSA)-targeting nanobody (JS014) is in a phase I clinical trial. HSA is one of the most abundant proteins in serum and has a half-life of around 19 days which makes it a good candidate for half-life extension strategies [214]. This strategy resulted in the half-life of JS014 to be 10-fold of that of recombinant IL-21 [215]. The construct showed strong antitumor efficacy in murine MC38 tumors. In addition, JS014 increased the antitumor effect of PD-1 and T cell immunoglobulin and ITIM domain (TIGIT) blockades in MC38 tumors [215,216].

A different construct, still in pre-clinical development, is an EGFR-targeting antibody (Erbitux) carrying IL-21 [217]. The construct was compared to anti-EGFR-IL-2 and showed much lower toxicity and improved antitumor efficacy in MC38 tumors bearing a chimeric EGFR, in which six amino acids of mouse EGFR were replaced by the residues found in human EGFR. The efficacy was achieved by expanding existing intra-tumoral functional CD8+ T cells rather than recruiting new lymphocytes. Additionally, EGFR-IL-21 could overcome checkpoint blockade resistance in mice with advanced MC38 tumors [217].

IL-21 was also fused to the fibronectin-targeting antibody F8 in a single-chain diabody format at the C-terminus [218]. Biodistribution experiments showed that this construct did not preferentially accumulate in the tumor site due to the peripheral binding of IL-21, highlighing the need to engineer IL-21 to improve its binding and activity at the tumor site.

In line with this concept, another IL-21-carrying construct is PD-1-targeted (AMG256) and was in a phase I trial in 2020 to evaluate its safety, tolerability, pharmacokinetics, and pharmacodynamics in patients with advanced solid tumors [219]. The rationale behind the construct was to reactivate the intra-tumoral CD8+ T cells by targeting them using anti-PD-1. The AMG256 construct does this by delivering a highly attenuated IL-21 mutant that was created by using structure-guided protein engineering. The mutant cytokine was designed to bind its receptor only when the antibody is already bound to PD-1 [220]. This design was used to restrict the activity of the cytokine to target cells and thereby improve the safety profile properties of the drug [220]. The fusion protein inhibited cancer growth in a humanized melanoma model compared to PD-1 treatment [220].

IL-21 was also fused to hu14.18. The resulting construct hu14.18–IL-21 ICK stimulated CD8+ T cells and M1 tumor-associated macrophages, decreased the recruitment of immunosuppressive cells to the TME, and mediated potent antitumor cytotoxicity against neuroblastoma more efficiently than hu14.18–IL-2 [136].

## 9. IL-10

IL-10 is recognized as an anti-inflammatory cytokine that operates as a dimer [1]. Its primary functions include the downregulation of antigen presentation and the inhibition of the release of pro-inflammatory mediators [1]. The role of IL-10 in cancer remains a subject of controversy. IL-10 is known to exert inhibitory effects on macrophages and the pro-inflammatory CD4+ Th17 T cell response [221]. Notably, mice and humans deficient in IL-10 display spontaneous development of inflammatory bowel disease [222]. The anti-inflammatory properties foster the pro-tumor image of IL-10. Nonetheless, individuals in whom proper IL-10 signaling is compromised, whether mice or humans, display a markedly heightened susceptibility to spontaneous tumor development [223,224]. In contrast, IL-10 treatment has surprisingly demonstrated antitumor efficacy, as shown with PEGylated IL-10, which induced the expression of IFNγ and granzymes in the tumor and induced the CD8+ T cell-mediated rejection of breast cancer models in mice. When challenged after 8 months, the mice rejected the tumor, suggesting durable immune memory induced by the treatment [225]. The immunostimulatory functions of the PEGylated form of IL-10 (pegilodecakin), that extend the cytokine lifetime, were confirmed in cancer patients [226], but it did not improve the outcome of patients with gemcitabine-refractory metastatic pancreatic [227] or metastatic lung cancer [228]. This led to the discontinuation of the pegilodecakin program, despite evidence of IL-10 biological effect in peripheral blood.

Due to its unexpected and relatively lately uncovered antitumor activity, IL-10 has also recently begun to receive attention in the ICK field. The F8 antibody carrying IL-10 is clinically being tested for the treatment of rheumatoid arthritis or ulcerative colitis [229] and would be an interesting construct to assess for cancer therapy. To date, there have been no ICKs incorporating IL-10 that have advanced to clinical trials for cancer treatment.

CmAb-(IL-10) is an anti-EGFR antibody (cetuximab)-based IL-10 ICK which displayed enhanced antitumor responses on an EGFR-positive tumor mouse model associated with a reduced toxicity. The authors draw the conclusion that this construct effectively hinders dendritic cell-mediated apoptosis of CD8+ T cells [230].

The anti-CSF1R-IL-10 is a fusion protein consisting of a murine immunoglobulin G2a (IgG2a) isotype anti-mouse colony-stimulating factor-1 receptor (CSF1R) antibody fused to the heavy-chain C-terminus via a linker sequence fused to IL-10. This construct intends to better deliver IL-10 in tumor-associated macrophages (TAMs) since CSF1R is highly expressed in TAMs. Treatment with this ICK demonstrated a reduction in TAMs and significant antitumor activity in preclinical models [231].

## 10. GM-CSF

Granulocyte-macrophage colony-stimulating factor (GM-CSF) is a glycoprotein secreted by immune cells, endothelial cells, and fibroblasts which drives the generation of myeloid cell subsets including monocytes, neutrophils, macrophages, and DCs in response to infections and cancers. The recombinant form of GM-CSF (sargramostim) has potent immunostimulant functions and is primarily used for myeloid reconstitution after bone marrow transplantation. In cancer, GM-CSF has been considered as a double-edged sword in the field of immunotherapy as it can, depending on the context, polarize TAMs toward M1 or M2 phenotypes, induce inflammatory neutrophils but enhance immunosuppressive MDSCs, increase pro-inflammatory DCs but also render them tolerogenic, and, finally, stimulate both effector T cells and Tregs [232]. Administration of GM-CSF has shown to provide clinical benefit in patients with several cancers [233,234,235,236]. In the field of anti-cancer vaccines, although GM-CSF has been widely used, it is unclear to which extent it contributed to the efficacy of treatment in these studies [237,238,239]. A critical factor influencing whether GM-CSF exerts anti- or pro-tumorigenic effects is the administrated dose [240] and the administration route. For example, the MDSC immunosuppressive effect of GM-CSF is seen only during its systemic administration but not when GM-CSF secretion is localized to the TME [241]. Therefore, targeted delivery of GM-CSF to the TME may be more beneficial than its systemic administration to promote DC maturation and enhance the presentation of tumor antigens.

Notably, anti-HER2/neu IgG3-(GM-CSF) is a full IgG ICK fused to GM-CSF that accumulated in the tumor and spleen in animal models and was able to enhance both CD4+ Th1- and Th2-mediated immune responses and reduce tumor growth in anti-HER2-resistant tumors [242].

More recently, interesting studies with the fusion protein rlipoE7m-MoGM were published. rlipoE7m-MoGM is a bifunctional protein which combines Toll-like receptor 2 (TLR2) agonist and GM-CSF activities. TLR members are expressed in myeloid cells such as DCs and are known to stimulate antigen presentation [243]. rlipoE7m is a recombinant lipoprotein engineered with a mutated HPV16 E7 tumor antigen that binds to TLR2 and can activate DCs and eradicate tumor progression in mice [244]. rlipoE7m-MoGM demonstrated strong antitumor immune response, both local and systemic, in mouse models [245].

## 11. IFNγ

IFNγ is a key immunoregulatory pleotropic cytokine that belongs to the type II class of interferons. IFNγ is secreted by immune cells themselves and dimerizes to induce strong innate and adaptive immune responses through the binding to IFNGR1 and IFNGR2. Like GM-CSF, IFNγ has a dual role in cancer as it can favor the antitumor immune response through ischemia induction, antigen presentation cell activation, macrophage repolarization, induction of Tregs fragility, and MDSC inhibition, but it also favors tumor escape via the induction of tolerant molecules expression (such as CTLA-4 and PD-L1), stimulating angiogenesis and promoting tumor progression [246].

In the clinic, administration of recombinant IFNγ (IFNγ1b, Actimmune) in combination with chemotherapy has shown controversial results [247,248,249]. Despite a high number of ongoing clinical trials, this cytokine has not been approved by the FDA to treat patients with cancers, except malignant osteoporosis, possibly because of the contribution of IFNγ to tumor evasion and the induction of adverse events [249].

Surprisingly, IFNγ has only attracted minor interest in the field of ICKs so far, likely because this cytokine is prone to an important “cytokine sink” effect due to its cognate receptors found in different organs which may limit its accumulation in the TME. IFNγ was fused to L19 and F8 antibodies as described above. The tumor targeting ability of the L19-IFNγ ICK was influenced by the number of IFNγ receptors expressed in the mouse and the F8-IFNγ ICK did not localized preferentially in the tumor as it could have been expected [250,251]. One recently published construct with improved tumor-homing properties is L19-IFNγ KRG. This ICK is an IgG4 format of the L19 antibody fused at the heavy-chain C-terminal domain to a truncated version of IFNγ designed to be protected against proteolytic digestion at its C-terminus and to show reduced affinity to its receptor [252]. L19-IFNγ KRG selectively localized to neoplastic lesions, led to tumor growth inhibition, and increased the intra-tumoral T and NK cells in combination with anti-PD-1.

chTNT-3/muIFNγ is another IFNγ-based ICK designed with the necrosis-targeting antibody chTNT-3. This ICK showed significant intra-tumoral retention and was able to decreased the number of metastatic tumors in an animal model without causing any observable toxicity, but its activity was not associated with any modulation of tumor-infiltrating cytotoxic T cells [253].

## 12. IFNα

Interferon alfa 2 (IFNα) is the most extensively characterized cytokine of the IFN type I family. IFNα modulates both innate and adaptive immune responses via activating monocytes and antigen-presenting cells. IFNα is used to treat several types of cancer, including hairy cell leukemia, melanoma, and renal cell carcinoma [254]. Mechanistically, IFNα2 exhibits direct cytotoxicity and anti-proliferative activities on tumor cells [255]. In addition, IFNα stimulates the antitumor immune function by activating dendritic cells [256]. However, the frequent administration of IFNα required to compensate for its short half-life produces toxic side effects including severe depression, fever, and headache leading to the abandonment of the treatment in many patients [257].

IFNα fused to rituximab demonstrated efficacy in lymphoma models though a molecular mechanism involving the binding of the construct to CD20 and the expression of IFNα receptor on tumor cells, suggesting a direct antitumor effect of this construct. However, the activity of this construct on the immune system was not described [258].

JZA01 and JZA02 are two IFNα-based ICKs in an Ig format fused to an anti-VEGFR2 designed to target solid tumors. JZA01 showed efficacy in tumor-bearing mice despite its reduced IFNα activity observed in vitro compared to native IFNα [259]. JZA02 has the same design as JZA01 but carries a mutated version of IFNα aiming to optimize affinity to its receptor which was reduced when fused as an ICK. Treatment with this construct showed improved antitumor efficacy compared to JZA01 and demonstrated enhanced dendritic cell activity and increased CD8+ T cell infiltration in the TME [260].

## 13. Summary and Perspectives

The cytokine field has expanded significantly over the last few decades, but their therapeutic use has faced several challenges, as described in this review. The growing number of published ICKs entering the clinical phase demonstrates the relevance of this technology, which provides several advantages and opportunities for improvement.

ICKs often increase the cytokine half-life, which was one of the main issues observed with naked cytokine administration. Depending on the format choice, ICKs offer the possibility to modulate tumor penetration and engage, or not, the binding to selected Fc receptors. This modulation affects pharmacokinetics and the contribution of antibody-mediated effector functions, which can impact the antitumor immune response and the extent of side effects simultaneously. The first ICKs were designed to only deliver cytokines to the TME. The advances in cancer biology understanding, associated with the discovery of therapeutic mAb, having antitumor activity on their own, enabled the generation of ICKs that combine several modes of action. The antibody moieties of ICKs have targeted a broad range of tumor-associated/specific antigens, initially mostly expressed by tumor cells, the extracellular matrix, and tumor-associated vessels. With the recent breakthrough of cancer immunotherapy, an increasing number of ICKs were designed to target immune cells, enabling, for example, the combination of immunostimulatory modes of action of the selected cytokine with immune checkpoint blockades. Although very promising and despite many completed clinical trials, none of the developed ICKs have been approved for use in the treatment of cancer so far [261] and there are still several ways of improvement to be considered before these molecules can translate into clinical use.

In many cases, the adverse effects linked to the use of the ICK format resemble those observed with free cytokines, and this is because the cytokine is selected for its ability to retain functionality when fused to an antibody. A possible future direction for ICKs is tumor-activatable cytokines. This solution would reduce the systemic activity of the cytokines and thereby most of their side effects. Activation mechanisms of a cap have been explored using enzymes that are highly expressed in the TME compared to healthy tissue. Such a construct could combine a cytokine capping or activation system in combination with an ICK [262]. Tumor-activatable cytokines can be achieved by using amino acid sequences, sensitive to tumor-enriched proteases, to fuse the cytokine with an mAb. This emerging approach was successful in preclinical models with the LH05 ICK (anti-PD-L1-IL-15) [160]. In addition, ICK fusions such as the anti-VEGFR2-INFα can exhibit a partial reduction in cytokine activity [259]. This might not only be a result of the nature of the cytokine but also of the cytokine position on the antibody or the type of fusion, which constitutes another way, potentially combinable with a tumor-activatable technology, to attenuate the cytokine and minimize both the peripheric “cytokine sink” effect and the cytokine systemic toxicity. Cytokine engineering is also an option to attenuate cytokine functions, as shown with the L19-TNF^I97A^ [171]. Activity-on-Target cytokines (AcTakines) are ICKs combining a targeting antibody and an engineered cytokine mutant with reduced receptor affinity. In AcTakines, the wild-type cytokine is replaced by a mutant version with strongly reduced affinity for its receptor. Consequently, they are inactive until they regain full activity on the targeted cells by local avidity-driven receptor binding [263]. Although this technology can reduce systemic toxicity, such constructs contain foreign epitopes that can induce immunogenicity, as reported in a phase I clinical trial with the Hu14.18-IL-2 ICK [264]. Nevertheless, the number of ICKs using the concept is progressing in clinical trials, demonstrating the promise of this therapeutic approach.

For ICK monotherapy that did not lead to the expected clinical efficacy, multiple cytokine-carrier ICKs may be considered. While Tripokin, a fully-human immunostimulatory product consisting of a trimeric TNF/IL-2/L19 ICK seems promising [265], an mDCH fusion protein, composed of an anti-epithelial cell adhesion molecule (EpCam) fused to IL-2 and GM-CSF, did not significantly synergize in tumor rejection [266]. The number of dual-cytokine ICKs is still limited but the scale and the variety of the cytokine repertoire open the path to many possible combinations which, if rationally selected, may have synergetic effects in vivo.

Finally, the increasing number of bispecific antibodies developed opens the possibility to design ICKs with additional modes of action via the engagement of two antigens such as the bispecific PD-1-IL-2v and anti-PD-L1 ICK [115]. The design of bispecific-antibody ICKs can also enable to further stimulate the antitumor immune response through the targeting of two cell types, similarly to the bispecific T cell engagers (BiTEs) [267].

In conclusion, multiple solutions are being developed around the ICK concept to fully optimize the therapeutic potential of cytokines. ICKs have not been approved yet, but the constantly evolving protein engineering technologies enabling a tumor-specific activation of the cytokines, together with the right selection of optimal combinations of cytokines with antibodies targeting tumor immunosuppressive mechanisms, might have synergic functional properties on the anti-tumor immune system which should translate to novel therapeutic options for patients with cancer.

## Data Availability

No new data were created or analyzed in this study. Data sharing is not applicable to this article.

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
