# Peer review of "Stimulating the Antitumor Immune Response Using Immunocytokines: A Preclinical and Clinical Overview"

_pharmaceutics, 2024, doi:10.3390/pharmaceutics16080974_

Round 1
Reviewer 1 Report
Comments and Suggestions for Authors
The authors described in this review the state of the art of the immunocytokines in a critical and explicative manner. This review could really help clarify the field pointing out the importance, complexity and number of research groups that are making an effort to bring immunocytokines to the market. The review is well written and well organized, only some small improvements are needed.
1) In describing the different formats of ICK maybe would be nice to add the “tandem diabody”, also called “single chain diabody”. Several cited ICK are in this format, like Dodekin et.
2) Line 280. HER2 is a TAA not TSA.
3) Figure 1. Would be possible to better divide the image representing on one side the pro-inflammatory and on the other side the anti-inflammatory with the corresponding cytokines releasing cells?
4) Line 514. About this study, I would mention the superiority of the F8-F8-SD-IL15 thanks to the Sushi Domain in defeating lung metastasis formation.
5) I would stress more the product Nidlegy (L19-TNF+L19-IL2) as one of the closest products close to reach the market authorization being in phase III for stage III B,C melanoma.
6) Line 572. Typo double “in”.
7) In vivo and in vitro should be written in italics.
8) About IL21, I remember an ICK using the F8 antibody that was not mentioned.
9) In the conclusion chapter, what is your opinion on the usage of mutated cytokines? Are you afraid of developing resistance against them?
Author Response
|
Comments 1: In describing the different formats of ICK maybe would be nice to add the “tandem diabody”, also called “single chain diabody”. Several cited ICK are in this format, like Dodekin et. |
|
Response 1: Thank you for pointing this out. We agree with this comment. Therefore, we have updated the main text and the legend of Figure 2 (lines 165 and 205/206). |
|
Comments 2: Line 280. HER2 is a TAA not TSA. |
|
Response 2: Agree. We have, accordingly, corrected TSA to TAA (line 300).
|
|
Comments 3: Figure 1. Would be possible to better divide the image representing on one side the pro-inflammatory and on the other side the anti-inflammatory with the corresponding cytokines releasing cells? Response 3: Agree. We have, accordingly, changed the figure 1 in the manuscript.
Comments 4: Line 514. About this study, I would mention the superiority of the F8-F8-SD-IL15 thanks to the Sushi Domain in defeating lung metastasis formation. Response 4: Agree. We have, accordingly, included this important aspect of this study in the main text (lines 514/515).
Comments 5: I would stress more the product Nidlegy (L19-TNF+L19-IL2) as one of the closest products close to reach the market authorization being in phase III for stage III B,C melanoma. Response 5: We have emphasized this point (line 601/602)
Comments 6: Line 572. Typo double “in”. Response 6: We have changed “in in vivo studies” to “in preclinical studies” to avoid a double use of “in” (line 573).
Comments 7: In vivo and in vitro should be written in italics. Response 7: Agree. We have corrected them all.
Comments 8: About IL21, I remember an ICK using the F8 antibody that was not mentioned.
Comments 9: In the conclusion chapter, what is your opinion on the usage of mutated cytokines? Are you afraid of developing resistance against them? Response 9: This is an important point, and we thank reviewer 1 for this question. We have included the risk of immunogenicity associated with the administration of mutated cytokines (lines 898-906). |
Reviewer 2 Report
Comments and Suggestions for Authors
Major points:
1. The authors mentioned that proinflammatory cytokines included IL-6, TNFα, IFNγ, and TGFβ (Page 2, line 61), while anti-inflammatory cytokines included IL-6 and IL-4 (Page 2, line 68-69). The information provided by the authors is inconsistent with Figure 1. In fact, TGF-β has both pro- and anti-inflammatory functions. Researchers should provide clearer illustrations to avoid confusion regarding these roles. Meanwhile, in the sections of immunocytokines (ICK), why ignore to discuss TGF-β?
2. In both the introduction section and Figure 1, the authors should provide a description of the roles played by IL-2, IL-12, IL-15, IL-21, IFNα and GM-CSF within the TME.
Minor points:
1. INFγ in page 1 (line 24) and page 3 (line 115) should be IFNγ. Please correct it.
2. When the term ‘Transforming Growth Factor’ (TGF) is first introduced in the document on page 2 (line 61), it should be spelled out in full along with its abbreviation. However, in subsequent content (page 12, line 528), only the abbreviation ‘TGF’ is necessary.
3. The sentence in the conclusion section " Even though ICK have not been approved yet, their potential remains immense, particularly based on the constantly...." (page 19, line 887-889) needs to be rewritten for better clarity. The Conclusion needs to be rewritten.
Author Response
|
Major points Comments 1: The authors mentioned that proinflammatory cytokines included IL-6, TNFα, IFNγ, and TGFβ (Page 2, line 61), while anti-inflammatory cytokines included IL-6 and IL-4 (Page 2, line 68-69). The information provided by the authors is inconsistent with Figure 1. In fact, TGF-β has both pro- and anti-inflammatory functions. Researchers should provide clearer illustrations to avoid confusion regarding these roles. Meanwhile, in the sections of immunocytokines (ICK), why ignore to discuss TGF-β? |
|
Response 1: Thank you for pointing this out. We agree with this comment. Therefore, we have added complementary information in the introduction regarding TGF-β (lines 77-80). In addition, we changed the organization and legend of Figure 1 to avoid any confusion (lines 123-140).
|
|
Comments 2: In both the introduction section and Figure 1, the authors should provide a description of the roles played by IL-2, IL-12, IL-15, IL-21, IFNα and GM-CSF within the TME. Response 2: Agree. We have, accordingly, changed the figure 1 and described the role of these cytokines in more details in the legend (lines 123-140).
|
|
Minor points
Comments 1: INFγ in page 1 (line 24) and page 3 (line 115) should be IFNγ. Please correct it |
|
Response 1. Thank you for pointing this out. We have corrected these typos (lines 24 and 118).
|
|
Comments 2: When the term ‘Transforming Growth Factor’ (TGF) is first introduced in the document on page 2 (line 61), it should be spelled out in full along with its abbreviation. However, in subsequent content (page 12, line 528), only the abbreviation ‘TGF’ is necessary. |
|
Response 2: Agree. We have modified this (lines 77-80 and 551).
|
|
Comments 3: The sentence in the conclusion section " Even though ICK have not been approved yet, their potential remains immense, particularly based on the constantly...." (page 19, line 887-889) needs to be rewritten for better clarity. The Conclusion needs to be rewritten. |
|
Response 3: Agree. We have, revised the conclusion (line 920-926).
|
Round 2
Reviewer 2 Report
Comments and Suggestions for Authors
The authors have answered all the questions.